# Tests of Charge–Parity Symmetry and Lepton Flavor Conservation in the Top Quark Sector

**Kai-Feng Chen** [1,*,†] **and Reza Goldouzian** [2,†]

1   Department of Physics, National Taiwan University, Taipei 10617, Taiwan
2   Department of Physics 225, Nieuwland Science Hall University of Notre Dame, Notre Dame, IN 46556-5670, USA
*   Correspondence: kai-feng.chen@cern.ch
†   These authors contributed equally to this work.

**Abstract:** The Standard Model (SM) of particle physics is the most general renormalizable theory which is built on a few general principles and fundamental symmetries with the given particle content. However, multiple symmetries are not built into the model and are simply consequences of renormalizabilty, gauge invariance, and particle content of the theory. It is crucial to test the validity of these types of symmetries and related conservation laws experimentally. The CERN LHC provides the highest sensitivity for testing the SM symmetries at high energy scales involving heavy particles such as the top quark. In this article, we are going to review the recent experimental searches of charge–parity and charged-lepton flavor violation in the top quark sector.

**Keywords:** charge–parity violation; charged-lepton flavor violation; top quark

## 1. Introduction

The Standard Model (SM) of particle physics is an extremely successful theory that has been extensively verified against experimental results. However, it can not explain several fundamental aspects of nature such as neutrino masses, dark matter, and baryon asymmetry in the universe. The SM Lagrangian is built based on a few principles including gauge symmetries and renormalizability. While the gauge symmetry is considered as fundamental rules in the SM Lagrangian, there are some symmetries in the SM that come from the structure imposed by the fundamental rules, called global symmetries. These symmetries can be exact or approximate and are very powerful tools for testing the SM validity. To explain the SM open issues, many models beyond-the-SM (BSM) have been proposed. These BSM models usually introduce new particles and interactions which lead to violation of the SM global symmetries. Therefore, testing the SM global symmetries is also considered as a unique window for probing BSM physics scenarios.

The combined transformation of parity (*P*) and charge-conjugation (*C*) operations was believed to be exact symmetry, until the discovery of *CP* violation in the neutral Kaon system in 1964 [1]. A model proposed by Kobayashi and Maskawa [2], denoted as the KM model, pointed out that an irreducible complex phase in the quark mixing matrix, known as the Cabibbo–Kobayashi–Maskawa (CKM) matrix, would enable *CP* violation in a natural manner, if there are at least three generations of quarks. With the discovery of the charm, bottom, and top quarks, together with the discovery of *CP* violation in the B-meson system [3,4], the KM model has been confirmed and became one of the key ingredients of the modern SM. However, even with the big success of the KM model, the known *CP* asymmetry in the SM is still far from enough to form a matter-dominant universe, hence it is necessary to look for additional sources of *CP* violation experimentally.

Top quark is playing an important role in the B-meson mixing, which is one of the main processes that *CP* violation can actually occur in the bottom sector. Unlike the bottom quark, the top quark itself has a rather monotonic flavor structure. Top always decays to

bottom and W-boson since $|V_{tb}|$ is much larger than $|V_{td}|$ and $|V_{ts}|$. As top is much heavier than any other quark, the GIM mechanism [5] is very effective and the flavor-changing neutral current processes are very suppressed. The $CP$ violation effect vanishes in the limit of $m_d, m_s \approx 0$; no other interfered process generates a non-zero strong phase to have measurable $CP$ asymmetries, which require the presence of both strong and weak phases.

However, if there are contributions from new physics, the situation can be very different. Any hint of $CP$ violation in the top sector will be a smoking gun signal for new physics [6]. With more than $10^8$ pairs of top quarks produced at the LHC, the experiments have a very good chance to probe sub-percent effects already. However, as the over-dominant $t \rightarrow bW$ decay, the usual measurements which involve different decays as introduced for strange and bottom quarks cannot be repeated for top. Alternative observables have to be introduced instead.

In the SM with massless neutrinos, the mixing of neutrino flavors is forbidden. Consequently, the flavor of charged leptons is conserved and couplings of leptons to gauge bosons are lepton flavor universal (LFU). The lepton flavor conservation in the neutral-lepton sector was found to be violated after the discovery of neutrino oscillations. Neutrino oscillations could also give rise to charged-lepton flavor violating (CLFV) processes. Because of the smallness of the neutrino masses, these processes are highly suppressed and are far below experimental sensitivity. Although the SM predictions for the CLFV processes are below experimental sensitivity, many theoretical scenarios beyond the SM, such as the two-Higgs doublet model [7], the minimal supersymmetric model [8], and the inverse seesaw model [9], predict detectable CLFV rates. Any evidence for such rare processes would therefore serve as a clear signature of physics beyond the SM.

The hints of LFU violations have been reported in semileptonic B decays, and the experimental evidence has risen over the past few years [10,11]. It is known that models accommodating violation of lepton universality generally also lead to observable effects in lepton flavor violation [12]. Models that can describe these small deviations also predict measurable observables in the top quark sector [13]. For example, certain leptoquark models that can accommodate the observed deviation in the B sector would imply branching fractions of $t \rightarrow \ell\ell'c \approx 10^{-6}$, with $\ell$ and $\ell'$ representing different-flavor charged leptons. Searching for CLFV processes related to the top quark would be complimentary to the searches in the B meson sector [14].

In this article, we discuss the measurements that have been carried out by ATLAS and CMS experiments in the context of $CP$ violation in the top quark sector. In addition, we review the results of the first search for the top quark CLFV interactions performed by the CMS collaboration.

## 2. Search for $CP$ Violation in Single Top Events

ATLAS has presented a note on the search for $CP$ violation in the lepton plus jets decay of single top quarks [15]. The top quarks are produced via t-channel where the top quarks are highly polarized, and it is possible to define CP-violating sensitive observables with the angles derived by the top decay products. The top quarks are exclusively decaying to W boson and bottom quark, and the Wtb vertex in the general effective operator framework can be expressed as [16,17]

$$\mathcal{L}_{Wtb} = -\frac{g}{\sqrt{2}}\bar{b}\gamma^{\mu}(V_L P_L + V_R P_R)tW_{\mu}^{-} - \frac{g}{\sqrt{2}}\bar{b}\frac{i\sigma^{\mu\nu}q_{\nu}}{m_W}(g_L P_L + g_R P_R)tW_{\mu}^{-} + \text{h.c.}, \quad (1)$$

where the weak coupling constant is denoted by $g$; $m_W$ and $q_\nu$ are the mass and four momentum of the W boson. The left (right) handed projection operators, vector, and tensor coupling are given by $P_L$ ($P_R$), $V_L$ ($V_R$), and $g_L$ ($g_R$), respectively. In the SM, the $V_L$ is simply the CKM element $V_{tb}$, and the anomalous couplings, $V_R$, $g_L$, and $g_R$ all vanish to zero.

In order to probe these couplings, it is possible to construct angular asymmetry observables based on the decay products of the top quark, such as a forward-backward asymmetry,

$$A_{FB} = \frac{N_{\text{evt}}(\cos\theta > 0) - N_{\text{evt}}(\cos\theta < 0)}{N_{\text{evt}}(\cos\theta > 0) + N_{\text{evt}}(\cos\theta < 0)}, \tag{2}$$

where $N_{\text{evt}}$ is the number of events with the criterion $\cos\theta > 0$ or $\cos\theta < 0$, and $\theta$ is an angle formed by the direction of top decay daughters. A typical choice of the angle (denoted as $\theta^*$) could be the angle between the direction of the lepton from the W decay in the W boson rest frame and direction of the W boson in the top quark rest frame. However, the asymmetry defined with $\theta^*$ is not really sensitive to the anomalous couplings related to *CP* violating complex phases.

As suggested in Refs. [18,19], in the t-channel single top production, the top quarks are mostly polarisation in the direction of the spectator quark; hence, by defining a new reference direction, $\overrightarrow{N} = \overrightarrow{s_t} \times \overrightarrow{q}$, where $\overrightarrow{s_t}$ is the direction of the spectator quark, and $\overrightarrow{q}$ is the momentum of W boson, all in the rest frame of the top quark. A new angle $\theta^N$ can be defined by the angle between the lepton in the W boson rest frame and the new direction $\overrightarrow{N}$. The forward-backward asymmetry defined by $\theta^N$, denoted as $A_{FB}^N$, could provide information of the anomalous coupling $g_R$, in particular for the imaginary part which is sensitive to the *CP* violation. In the SM, the coupling $g_R$ is very close to zero, $(-7.17 - 1.23i) \times 10^{-3}$ [20].

ATLAS has performed a measurement of $A_{FB}^N$ and provided a bound on the imaginary part of $g_R$ based on the data collected at $\sqrt{s} = 7$ TeV, corresponding to an integrated luminosity of 4.66 fb$^{-1}$. The data events are required to have at least an electron or a muon at the trigger. At the analysis level, an offline reconstructed electron or muon, significant transverse missing energy $E_T^{\text{miss}}$, and two jets (one of the jets must be tagged as originated from b-quark) are required for the signal candidates. The candidate electron should have a transverse momentum $p_T$ greater than 25 GeV, and within $|\eta| < 2.47$ (while the transition region $1.37 < |\eta| < 1.52$ is excluded). For the candidate muon, the minimal $p_T$ is also 25 GeV and should be within the region $|\eta| < 2.5$. Jets are required to have a minimal $p_T$ greater than 30 GeV; $E_T^{\text{miss}}$ and the transverse mass of the reconstructed W boson should be above 30 GeV as well.

The top quark is reconstructed from the b-jet and the W boson, where the W boson is reconstructed with the charged lepton and neutrino momenta. The transverse momentum of neutrino is assumed to be the same as $E_T^{\text{miss}}$, while the longitudinal component is solved by the constraint of W boson mass. In the case of two possible solutions, the solution with smaller longitudinal neutrino momentum is selected. Based on the reconstructed momenta of top quark and W boson, the decay angle $\theta^N$ can be derived. The distribution of $\cos\theta^N$ includes the detector effects and has to be unfolded back to the parton level. The forward-backward asymmetry $A_{FB}^N$ is then computed based on the unfolded distribution of $\cos\theta^N$:

$$A_{FB}^N = 0.031 \pm 0.065\,(\text{stat.})_{-0.031}^{+0.029}\,(\text{syst.}). \tag{3}$$

The major systematic uncertainties considered in the measurement of $A_{FB}^N$ are: signal t-channel single top modeling, $t\bar{t}$ modeling, background normalization, jet energy scale and resolution, lepton selection and trigger. The effects of rest systematic sources are all smaller than 0.005.

The relation between $A_{FB}^N$ and $g_R$ can be approximated by $A_{FB}^N = 0.64 P\Im(g_R)$ [21] if the values of $g_R$ are small (assuming $V_L = 1$, $V_R = 0$, and $g_L = 0$). Assuming a value of $P = 0.9$ derived from the knowledge of top quark polarization, the limit on the value of $\Im(g_R)$ is obtained to be $[-0.20, 0.30]$ at the 95% confidence level. The measured $A_{FB}^N$ and the derived limit on $\Im(g_R)$ are both consistent with the predictions from the SM, and no significant *CP*-violating effect is observed in the t-channel single top production.

## 3. Measurements of *T*-Odd Triple-Product Observables

A triple-product correlation is constructed with three directional vectors and takes the form of $\vec{v_1} \cdot (\vec{v_2} \times \vec{v_3})$. The vectors could be particle momenta or spins. The combined triple product is odd under time-reversal (T) transformation; under the CPT theorem, a T-odd observable is also a CP-odd observable. In the $pp \to t\bar{t} \to b\bar{b}W^+W^-$ production and decay chain, there are multiple momentum vectors which can be reconstructed experimentally, but how to select useful combinations of triple-products is a non-trivial task. The authors of Refs. [22,23] examined a series of triple-product observables and studied the contributions from a chromoelectric dipole moment (CEDM) of the top quark. The magnetic and electric couplings between the top quarks and gluons can be expressed as

$$\mathcal{L} = \frac{g_s}{2} \bar{t} T^a \sigma^{\mu\nu} (a_t^g + i\gamma_5 d_t^g) t G_{\mu\nu}^a, \tag{4}$$

where the strong coupling constant and the gluon field strength tensor are denoted by $g_s$ and $G_{\mu\nu}^a$, respectively; the parameters $a_t^g$ and $d_t^g$ are for the chromomagnetic and chromoelectric dipole moments. The term $d_t^g$ can be further written as $d_t^g = \frac{\sqrt{2}v}{\Lambda^2}\Im(d_{tG})$, where $\Lambda$ is the scale of the BSM phenomena, $v \approx 246$ GeV is the vacuum expectation value, and $d_{tG}$ is the CEDM parameter. CMS has performed several studies along this proposal, including dilepton channel [24] and lepton plus jets decays [25,26].

### 3.1. Search in the Dilepton Channel

The analysis in the dilepton channel exploits the top-pair production, with both W bosons decaying leptonically to define two CP-odd correlations:

$$\mathcal{O}_1 = \epsilon(p_t, p_{\bar{t}}, p_{\ell^+}, p_{\ell^-}), \tag{5}$$

$$\mathcal{O}_3 = \epsilon(p_b, p_{\bar{b}}, p_{\ell^+}, p_{\ell^-}), \tag{6}$$

where $p_t$ ($p_{\bar{t}}$), $p_b$ ($p_{\bar{b}}$), $p_{\ell^-}$ ($p_{\ell^+}$) are the four momenta of top (anti-top) quark, bottom (anti-bottom) quark, and lepton (anti-lepton), respectively, and $\epsilon$ is the Levi–Civita tensor. The CP violation can be tested by the measurements of the asymmetries,

$$A_{\mathcal{O}_i} = \frac{N(\mathcal{O}_i > 0) - N(\mathcal{O}_i < 0)}{N(\mathcal{O}_i > 0) + N(\mathcal{O}_i < 0)}. \tag{7}$$

These two asymmetries are the observables with the highest sensitivity and linear to $d_{tG}$.

The data used in the dilepton analysis are from pp collisions at 13 TeV correspond to an integrated luminosity of 35.9 fb$^{-1}$. The events are required to pass the single or dilepton trigger conditions, and then categorized according to the combination of lepton flavors, $e^+e^-$, $\mu^+\mu^-$, or $e^{\pm}\mu^{\mp}$. Electron candidates are reconstructed with CMS tracker and calorimeter information, must have a $p_T > 25$ (20) GeV for the leading (sub-leading) candidate, and are within the volume $|\eta| < 2.4$. Muon candidates are reconstructed with a combination of tracker and muon system information and fulfill the same criteria on the transverse momentum and pseudorapidity. The jets are reconstructed using the anti-$k_T$ algorithm with a distance parameter $R = 0.4$, and are required to have $p_T > 30$ GeV and $|\eta| < 2.4$. If a jet candidate is too close ($\Delta R = \sqrt{\Delta\eta^2 + \Delta\phi^2} < 0.4$) to a lepton candidate, the jet itself is removed.

An event is required to have two charged leptons, and at least two jets. One of the selected jets must be tagged as originating from bottom quark. The events with additional lepton of $p_T > 20$ GeV are discarded. The invariant mass of dilepton pairs must be greater than 20 GeV to suppress contributions from low-mass resonance decay and the Drell–Yan process. The same-flavor dilepton events with an invariant mass in the region between 76 and 106 GeV are also rejected to suppress the contributions from Z boson decays; however,

these events in the Z mass region are still used to normalize the Drell–Yan background contribution in the analysis.

The four momenta of quark and anti-quark used in the construction of observables are resolved by a reconstruction based on kinematic information: the transverse missing momentum is assumed to originate from the neutrinos, and the mass of reconstructed W boson and top quark should be equal to 80.4 GeV and 172.5 GeV, respectively. Detector resolution effects are taken into account through a smearing of the measured energies and directions of the reconstructed objects. The efficiency of this reconstruction procedure is around 90%; events without solutions for the neutrino momenta are excluded.

The observables are computed using the resolved four momenta of quark and anti-quark. A maximum likelihood fit is introduced to extract the asymmetries from the observables. The effects of systematic uncertainties are estimated by varying the nominal inputs and samples by the uncertainties, and repeating the full measurements. Most of the uncertainty sources are naturally cancelled in the asymmetries. The remaining major uncertainties are from limited simulated background samples, jet energy resolution and scale, top modeling such as color reconnection. None of the uncertainty source results have a variation larger than 0.003 on the asymmetries. The resulting asymmetries, which are summarized in Table 1, are found to be consistent with zero and used to derive a measurement on the CEDM parameter $d_{tG}$. The measured asymmetries and the coefficients of CEDM term are consistent with the expectation from the SM.

**Table 1.** The measured asymmetries $A_{\mathcal{O}_1}$ and $A_{\mathcal{O}_3}$, and the CEDM parameter $d_{tG}$.

| Observable | Asymmetry ($\times 10^{-3}$) | $d_{tG}$ |
|:---:|:---:|:---:|
| $\mathcal{O}_1$ | $2.4 \pm 2.8(\text{stat}) \pm 2.8(\text{syst})$ | $0.10 \pm 0.12(\text{stat}) \pm 0.12(\text{syst})$ |
| $\mathcal{O}_3$ | $0.4 \pm 2.8(\text{stat}) \pm 2.2(\text{syst})$ | $0.00 \pm 0.13(\text{stat}) \pm 0.10(\text{syst})$ |

*3.2. Analysis in the Lepton Plus Jets Channel*

The lepton plus jets analysis exploits a different set of observables recommended in Ref. [23]:

$$\mathcal{O}_3 = Q_\ell \epsilon(p_b, p_{\bar{b}}, p_\ell, p_{j_1}) \propto Q_\ell \vec{p}^{\,*}_b \cdot (\vec{p}^{\,*}_\ell \times \vec{p}^{\,*}_{j_1}), \tag{8}$$

$$\mathcal{O}_6 = Q_\ell \epsilon(P, p_b - p_{\bar{b}}, p_\ell, p_{j_1}) \propto Q_\ell(\vec{p}_b - \vec{p}_{\bar{b}}) \cdot (\vec{p}_\ell \times \vec{p}_{j_1}), \tag{9}$$

$$\mathcal{O}_{12} = q \cdot (p_b - p_{\bar{b}}) \epsilon(P, q, p_b, p_{\bar{b}}) \propto (\vec{p}_b - \vec{p}_{\bar{b}})_z \cdot (\vec{p}_b \times \vec{p}_{\bar{b}})_z, \tag{10}$$

$$\mathcal{O}_{14} = \epsilon(P, p_b + p_{\bar{b}}, p_\ell, p_{j_1}) \propto (\vec{p}_b + \vec{p}_{\bar{b}}) \cdot (\vec{p}_\ell \times \vec{p}_{j_1}). \tag{11}$$

The symbol $*$ indicates the momenta at the $b\bar{b}$ center-of-mass frame; the $z$ subscript indicates the projection along the beam axis; $P$ ($q$) is the sum (difference) of the four-momenta of the protons in the pp collision; $p_j 1$ represents the momentum of the jet with highest transverse momentum; $Q_\ell$ is the lepton charge. The presence of CP violation will result in a nonzero asymmetry defined in Equation (7).

The study uses the data collected at $\sqrt{s} = 13$ TeV corresponding to an integrated luminosity of 138 fb$^{-1}$. Trigger requirement includes the presence of an isolated lepton with a $p_T$ above 24–35 GeV. In the offline analysis, the electron candidates are required to have $p_T > 38$ GeV and within $|\eta| < 2.4$ (excluding the gap between barrel and endcap calorimeter, $1.44 < |\eta| < 1.57$). Muon candidates are required to have a transverse momentum above 30 GeV and within $|\eta| < 2.4$. The flavor of lepton (electron or muon) is used to categorizing the events too.

Jets are clustered using the anti-$k_T$ algorithm with a distance parameter of 0.4, with a minimal $p_T$ threshold of 30 GeV. Jets should be separated from the selected lepton candidate by an angular distance of $\Delta R > 0.4$. Jets from the hadronization of bottom quarks are identified using a deep-learning neural network based algorithm.

The events are required to have a reconstructed lepton, and at least four reconstructed jets; two of the jets should be tagged as bottom quarks. The association of top quark and anti-top quark with the final state jets and leptons are resolved with a $\chi^2$ algorithm that introduces the masses of top quark and W boson as constrained conditions: $\chi^2 = [(m_{jjb} - m_t)/\sigma_t]^2 + [(m_{jj} - m_W)/\sigma_W]^2$, where $m_{jjb}$ is the invariant mass of three jets (two non-b-tagged and one b-tagged); $m_t$, $\sigma_t$, $m_W$ and $\sigma_W$ are the mass of top quark, resolution of reconstructed top quark mass, mass of W boson, and resolution of reconstructed W boson mass, respectively. The object assignment is decided by selecting the permutation with the lowest $\chi^2$ score. By imposing the requirements of $\chi^2 < 20$ and $m_{lb} < 150$ GeV, the fraction of correctly assigned b jets is around 74% with an efficiency of 65% and a purity of $t\bar{t}$ events is 95%.

The measured asymmetry can be diluted by the detector and mis-reconstruction. In the analysis, the effect is parametrized with a dilution factor $D$, and the raw asymmetry (denoted as $A'_{CP}$) and the ideal $A_{CP}$ are related as a multiplicative correction $A'_{CP} = DA_{CP}$. The values of $D$ and the associated systematic uncertainties have been determined from simulations: $D(\mathcal{O}_3) = 0.46^{+0.01}_{-0.02}$, $D(\mathcal{O}_6) = 0.44^{+0.01}_{-0.02}$, $D(\mathcal{O}_1 2) = 0.74^{+0.01}_{-0.02}$, and $D(\mathcal{O}_1 4) = 0.60 \pm 0.01$. The observables $\mathcal{O}_3$ and $\mathcal{O}_6$ require distinguishing bottom and anti-bottom quarks, resulting in a lower chance of correct reconstruction and smaller $D$ value. Nevertheless, as $A_{CP}$ and $A'_{CP}$ are linearly dependent, a non-zero $A'_{CP}$ is already evidence of CP violation in the top sector; hence, the raw asymmetry $A'_{CP}$ is considered as the primary result of the analysis.

The asymmetries are computed with the signal yields with positive or negative values of $\mathcal{O}_i$. These yields are determined with an extended maximum likelihood fit to the invariant mass distributions of lepton and b-tagged jet. Systematic uncertainties are also largely cancelled in the asymmetry measurements. The detector effects are studied using an event-mixing method, which is mixing the momentum of the b-tagged jet and the highest $p_T$ light-flavor jet across different events. Other experimental and theoretical systematic uncertainties are all found to be tiny in this analysis, resulting in a total systematic uncertainty smaller than 0.001 on the asymmetries. The measured asymmetries can be used to provide a constraint on the CEDM contributions. The detector effects are first removed by dividing the dilution factors. The constraints on CEDM parameter are computed from the corrected asymmetries and then combined with the correlation among the CP observables taking into account. The resulting asymmetries as well as the derived CEDM parameters $d_{tG}$ are summarized in Table 2; the parameter $d_{tG}$ is measured to be $0.04 \pm 0.10 \pm 0.07$ with a combination of the results from the four observables. The measured asymmetries on the CP observables are consistent with the expectation from the SM, and show no hints for CP-violating effects.

**Table 2.** The measured raw $A'_{CP}$, the corrected asymmetry $A_{CP}$, and the derived CEDM parameter $d_{tG}$. The first uncertainty is statistical, and the second is systematic.

| Observable | $A'_{CP}$ (%) | $A_{CP}$ (%) | $d_{tG}$ |
|:---:|:---:|:---:|:---:|
| $\mathcal{O}_3$ | $-0.05 \pm 0.09^{+0.04}_{-0.07}$ | $-0.10 \pm 0.20 \pm 0.14$ | $+0.04 \pm 0.11 \pm 0.07$ |
| $\mathcal{O}_6$ | $-0.13 \pm 0.09^{+0.05}_{-0.07}$ | $-0.30 \pm 0.21 \pm 0.16$ | $+0.25 \pm 0.20 \pm 0.15$ |
| $\mathcal{O}_{12}$ | $+0.09 \pm 0.09^{+0.03}_{-0.05}$ | $+0.12 \pm 0.13 \pm 0.07$ | $+0.45 \pm 0.47 \pm 0.27$ |
| $\mathcal{O}_{14}$ | $-0.17 \pm 0.09^{+0.09}_{-0.02}$ | $-0.29 \pm 0.16 \pm 0.14$ | $-0.81 \pm 0.48 \pm 0.44$ |

## 4. Search for CLFV in Top Quark Production and Decay

A search for CLFV in both top quark associated production, and decay is presented by the CMS Collaboration [27]. The effective field theory approach is followed for parametrizing the CLFV effects. The top related operators are categorized based on their Lorentz structure to vector-, scalar- and tensor-like operators as the following:

$$O_{\text{vector}} = (\bar{l}_a \gamma^\mu l_b)(\bar{q}_c \gamma_\mu q_d) + (\bar{l}_a \gamma^\mu l_b)(\bar{u}_c \gamma_\mu u_d) + (\bar{e}_a \gamma^\mu e_b)(\bar{q}_c \gamma_\mu q_d) + (\bar{e}_a \gamma^\mu e_b)(\bar{u}_c \gamma_\mu u_d), \tag{12}$$

$$O_{\text{scalar}} = (\bar{l}_a e_b)\, \varepsilon\, (\bar{q}_c u_d) + \text{h.c}, \tag{13}$$

$$O_{\text{tensor}} = (\bar{l}_a \sigma^{\mu\nu} e_b)\, \varepsilon\, (\bar{q}_c \sigma_{\mu\nu} u_d) + \text{h.c}, \tag{14}$$

where $a \neq b$ are lepton-flavor indices, $c$ and $d$ are quark-flavor indices, q and l represent left-handed fermion doublets, u and e the right-handed fermion singlets, $\tau^I$ the Pauli matrices, $\varepsilon \equiv i\tau^2$ is the antisymmetric $SU(2)$ tensor, $\sigma^{\mu\nu} = \frac{i}{2}[\gamma^\mu, \gamma^\nu]$, and $\gamma^\mu$ the Dirac matrices. Three Wilson coefficients $C_{\text{vector}}$, $C_{\text{scalar}}$, and $C_{\text{tensor}}$ are probed individually in this analysis. The CLFV interactions contribute to the single top production and decay of the top quark in top quark–antiquark pair production ($t\bar{t}$). Due to the larger cross section of the CLFV process in the production mode compared to the decay mode and more distinctive kinematic distributions of the production mode with respect to the SM background, the production mode plays a leading role in the sensitivity of the search. This analysis presents results of the first search for "eμtu" and "eμtc" CLFV interactions in the eμ final state.

The analysis is based on pp collisions collected by the CMS detector at the LHC at a center-of-mass energy of 13 TeV, corresponding to an integrated luminosity of 138 fb$^{-1}$. Events with one oppositely charged electron–muon pair in the final state, along with at least one jet identified as originating from a bottom quark (b-tagged jet) are selected. The leading (sub-leading) lepton $p_T$ should be greater than 25(20) GeV and to lie within $|\eta| < 2.4$. Selected events are required to have at least one b-tagged jet with $p_T > 20$ GeV and $|\eta| < 2.4$. Selected background events dominated with SM $t\bar{t}$ events ($\approx$90%), followed by single top quark production in association with a W boson (tW) ($\approx$10%). The contributions from the SM background processes are estimated using the simulated events. Events are categorized further based on number of b-tagged jets to signal and $t\bar{t}$ control regions by requiring exactly one and greater than 1 b-tagged jets, respectively. A boosted decision tree (BDT) is trained based on the distinctive features of the signal process in the production mode with respect to the main backgrounds to maximize the sensitivity of the search.

Various sources of systematic uncertainty from modeling of the detector response and theoretical modeling of the signal and background processes are considered in this search. The final BDT distribution in the signal region $t\bar{t}$ control regions fitted simultaneously using a binned likelihood function to test for the presence of signal events. All the systematic uncertainties are treated as nuisance parameters in the fit. The fit results are consistent with the SM prediction and show no evidence for the presence of the CLFV signal. Therefore, upper limits are set on the signal cross sections at a 95% confidence level using the modified frequentist CLs method. Upper limits on the cross section of the CLFV processes are translated to the upper limits on the Wilson coefficients and related branching fractions of the top quark $\mathcal{B}(t \to e\mu q)$, q = u (c) quark. Limits obtained for vector-, scalar-, and tensor-like interactions are summarized in Table 3. In Figure 1, the results for two-dimensional limits on CLFV Wilson coefficients and branching fractions are displayed.

**Table 3.** Expected and observed 95% CL upper limits on the CLFV Wilson coefficients and top quark CLFV branching fractions.

| Vertex | Int. Type | $C_{e\mu tq}/\Lambda^2$ [TeV$^{-2}$] Exp | Obs | $\mathcal{B}(10^{-6})$ Exp | Obs |
|---|---|---|---|---|---|
| | Vector | 0.12 | 0.12 | 0.14 | 0.13 |
| eμtu | Scalar | 0.23 | 0.24 | 0.06 | 0.07 |
| | Tensor | 0.07 | 0.06 | 0.27 | 0.25 |

**Table 3.** *Cont.*

| Vertex | Int. Type | $C_{e\mu tq}/\Lambda^2$ [TeV$^{-2}$] | | $\mathcal{B}(10^{-6})$ | |
|--------|-----------|------|------|------|------|
| | | Exp | Obs | Exp | Obs |
| | Vector | 0.39 | 0.37 | 1.49 | 1.31 |
| eµtc | Scalar | 0.87 | 0.86 | 0.91 | 0.89 |
| | Tensor | 0.24 | 0.21 | 3.16 | 2.59 |

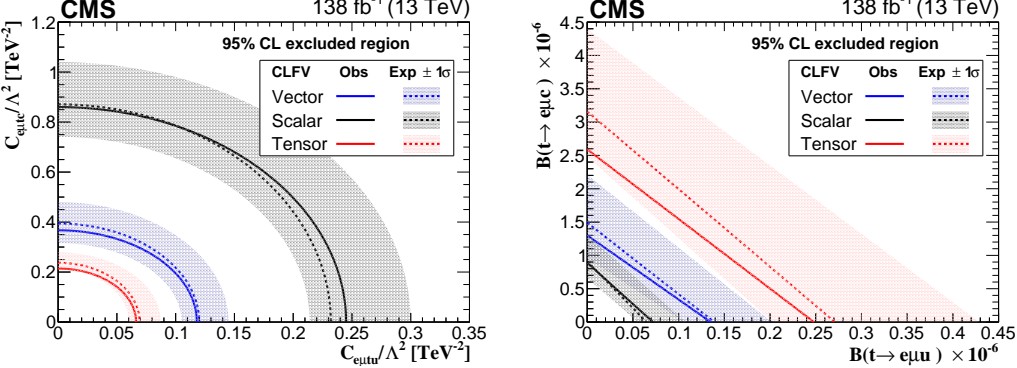

**Figure 1.** The observed 95% CL exclusion limits on the eµtc; of the eµtu Wilson coefficient (**left**) and $\mathcal{B}(t \to e\mu c)$ as a function of $\mathcal{B}(t \to e\mu u)$ (**right**) for the vector-, scalar-, and tensor-like CLFV interactions. The hatched bands indicate the regions containing 68% of the distribution of limits expected under the background-only hypothesis.

## 5. Conclusions and Outlook

In this article, several studies on the tests of charge–parity (CP) symmetry and lepton flavor conservation have been reviewed. ATLAS collaboration has performed a measurement of a CP-violating forward-backward asymmetry in t-channel single top quark events, where top quarks are expected to be highly polarized. This measurement is consistent with null forward-backward asymmetry, and CP is conserved. The result has been used to derive a constraint on the anomalous coupling under the effective operator framework. CMS collaboration has looked for CP asymmetries with top quark pair productions in dilepton and lepton plus jets' final states. The CP asymmetries are tested with the CP-odd triple-product observables, constructed using the four-momenta reconstructed from with the final-state particles. The asymmetries of the proposed observables are measured to be consistent with zero and then converted to the measurements of the chromoelectric dipole moment contribution to the top quark. A search for charged-lepton flavor violation (CLFV) has been carried out by CMS in top quark production and decay as well. Events with oppositely charged electron–muon pairs are selected in the study, and no significant excess over the background is observed. Limits are set on the decay branching fractions as well as the strength of four-fermion effective operators in the framework of an effective field theory approach.

Although none of the existing experimental searches to date shows evidence of symmetry breaking effects in the top sector, the studies should be still pursued. As the Standard Model of particle physics failed to explain several important aspects of the nature, such as baryogenesis of the universe and the origin of neutrino masses, substantial tests of these fundamental symmetries and the conservation laws governing them are the key methods to explore the open issues in the particle physics. As the top quark does play an important role in the Standard Model, it is mandatory to refine the existing measurements and look for other possibilities in the future. In particular, the upcoming high-luminosity LHC project will provide more than a factor of twenty statistics of top quarks, compared to the currently accumulated datasets during LHC Run-2. Precision of existing analyses can be further

improved by one or two orders of magnitudes. More observables can be further examined with the data produced at future LHC or with the planned high-energy $e^+e^-$ colliders.

**Author Contributions:** Writing—review and editing, K.-F.C. and R.G. All authors have read and agreed to the published version of the manuscript.

**Funding:** This research received no external funding.

**Institutional Review Board Statement:** Not applicable.

**Informed Consent Statement:** Not applicable.

**Data Availability Statement:** No new data were created or analyzed in this study. Data sharing is not applicable to this article.

**Conflicts of Interest:** The authors declare no conflict of interest.

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
