# Peer review of "Tests of Charge–Parity Symmetry and Lepton Flavor Conservation in the Top Quark Sector"

_universe, doi:10.3390/universe9020062_

Round 1
Reviewer 1 Report
Dear authors,
Thanks for this interesting and thorough review on "Tests of Charge-Parity Symmetry and Lepton Flavor Conservation in the top quark sector".
Few minor comments:
- Introduction: you could improve the flow of the text between lines 34-48.
- Line 62: "has been rising over the past few years" --> references?
- Line 132 --> eta (should be the greek letter)
- Line 274: "standard model" --> SM or Standard Model.
More important request:
The review deserves few more details on the prospects: HL-LHC and future e+e- colliders but alsoon the results still to come from LHC current and previous runs. For instance, ATLAS has only shown results for 4.7fb- 1?
Please extend this section with quantitative estimations.
If it is a matter of space, some details on object definition/cuts could be shortened, the references are clear enough on this if the reader is interested on them.
Regards
Author Response
Dear Reviewer,
Many thanks for your comments and suggestions. Please find our responses inline:
- Introduction: you could improve the flow of the text between lines 34-48.
- We have swap the senstences to improve the flow and connections between the two paragraphs.
- Line 62: "has been rising over the past few years" --> references?
Line 132 --> eta (should be the greek letter)
Line 274: "standard model" --> SM or Standard Model.- All fixed.
- The review deserves few more details on the prospects: HL-LHC and future e+e- colliders but alsoon the results still to come from LHC current and previous runs. For instance, ATLAS has only shown results for 4.7fb- 1? Please extend this section with quantitative estimations.
- ATLAS has only shown the analysis with 4.7 fb-1 so far. We wish to see the new results from the collaboration too.
- We agree it will be better to have more discussions and quantitative estimations for future colliders. Unfortunately it is not straightforward for us to provide proper/trustworthy estimations, as some of the key factors are already dominant by systematic uncertainties; in particular the principle of measurement may need to be refined for e+e- machine. Hence we decided to keep the current construction of the conclusion paragraph.
Reviewer 2 Report
This article presents a brief review of experimental constraints on CP violation in the top quark, especially as manifested by a nonzero top quark chromoelectric dipole moment. On the whole, the article is well-written and informative. It is useful to have such a summary of results from both ATLAS and CMS in different observables, and I recommend it for publication.
There are several places where the style/grammar needs to be fixed: sentence fragments (eg Line 61) or run-ons (Line 40) and typos (Line 132). I also have two points of straightforward clarification to request:
- In line 40, I don’t understand the sentence fragment, “no other process to generate a nonzero zero strong phase to have measurable CP asymmetries.”
- In line 104, is the numerical pre factor relating AFP to Im(gt) arising from carrying out the full cross section calculation? A comment or explanation would be helpful
- Line 117, what is dtG (as distinct from dtg the CEDM of the top)? It is referred to later in lines 209,210 and Table 2 as a “CEDM parameter”
Upon addressing these questions, the article can proceed to publication.
Author Response
Dear Reviewer,
Many thanks for your comments and suggestions. Please find our responses inline:
- There are several places where the style/grammar needs to be fixed: sentence fragments (eg Line 61) or run-ons (Line 40) and typos (Line 132).
- These issues have been fixed in the draft. Many thanks for spotting them!
- I also have two points of straightforward clarification to request:
In line 40, I don’t understand the sentence fragment, “no other process to generate a nonzero zero strong phase to have measurable CP asymmetries.”- This sentence has been extended to make it more clear.
- In line 104, is the numerical pre factor relating AFP to Im(gt) arising from carrying out the full cross section calculation? A comment or explanation would be helpful
- As the full comment can be lengthy, we decided to add an explicit citation (Aguilar-Saavedra and Bernabeu, Nucl. Phys. B 840, (2010) 349-378.) for this relationship.
- Line 117, what is dtG (as distinct from dtg the CEDM of the top)? It is referred to later in lines 209,210 and Table 2 as a “CEDM parameter”
- dtG is purely a parameter/coefficient in the model. We have changed text to avoid the confusion between dtG and dtg.